# Determining molecular properties with differential mobility spectrometry and machine learning

Stephen W.C. Walker[1], Ahdia Anwar[1], Jarrod M. Psutka[1], Jeff Crouse[1], Chang Liu [2], J.C.Yves Le Blanc[2], Justin Montgomery[3], Gilles H. Goetz [3], John S. Janiszewski[3], J. Larry Campbell[1,2] & W. Scott Hopkins[1]

The fast and accurate determination of molecular properties is highly desirable for many facets of chemical research, particularly in drug discovery where pre-clinical assays play an important role in paring down large sets of drug candidates. Here, we present the use of supervised machine learning to treat differential mobility spectrometry – mass spectrometry data for ten topological classes of drug candidates. We demonstrate that the gas-phase clustering behavior probed in our experiments can be used to predict the candidates' condensed phase molecular properties, such as cell permeability, solubility, polar surface area, and water/octanol distribution coefficient. All of these measurements are performed in minutes and require mere nanograms of each drug examined. Moreover, by tuning gas temperature within the differential mobility spectrometer, one can fine tune the extent of ion-solvent clustering to separate subtly different molecular geometries and to discriminate molecules of very similar physicochemical properties.

[1] Department of Chemistry, University of Waterloo, Waterloo N2L 3G1 ON, Canada. [2] SCIEX, 71 Four Valley Drive, Concord L4V 4V8 ON, Canada. [3] Pfizer Global Research and Development, Eastern Point Road, Groton 06340 CT, USA. Correspondence and requests for materials should be addressed to J.L.C. (email: Larry.Campbell@sciex.com) or to W.S.H. (email: Scott.Hopkins@uwaterloo.ca)

Although mass spectrometry-based techniques have long found utility for fast and sensitive compound detection and for gas-phase properties measurements (e.g., ion structure, reactivity), the efficacy of employing gas-phase measurements to predict condensed phase properties remains a contested subject[1,2]. However, differential mobility spectrometry (DMS) has recently shown promise in this regard[2–4]. Most commonly employed as a narrow-pass filter to separate ions from chemical noise[5,6], DMS takes advantage of an oscillating asymmetric electric field to drive rapid cycles of microsolvation and evaporation in environments that are seeded with low partial pressures of solvent vapor[3,7,8]. In examining how ion microsolvation properties are affected by specific structural attributes, the influence of electronic and resonance effects[2,9], the steric hindrance of charge sites[2,10,11], and the influence of intramolecular hydrogen bonding have been observed[12]. Thus, a picture is now emerging which suggests that the dynamic DMS environment facilitates swift and statistical sampling of an analyte's potential energy surface. Consequently, strong correlations may be observed between an analyte's clustering behavior within the DMS instrument and molecular properties in the condensed phase, such as cell permeability, which are partially dependent on the interaction potential between the analyte and its condensed phase environment[2,4,13–15].

During the early stages of drug development, it is the goal of medicinal chemists to relate the cellular availability of a drug candidate to its physicochemical properties and chemical structure[16,17]. Thus, from the perspective of rational drug design, a fast and sensitive technique that can aid in predicting molecular properties is highly desirable. At present, there are several benchtop in vitro techniques available to measure cellular permeability (e.g., RRCK, Caco-2, and MDCK)[18] and other physicochemical properties that serve as preliminary appraisals of in vivo behavior (e.g., turbidometry, PAMPA, LogD, $pK_a$)[19]. However, it is often challenging to discriminate between closely related structural analogues using the current state-of-the-art techniques[20,21]. Moreover, while exceptional progress has been made in refining these methods[22–24], they still require considerable resources and often experience issues with reproducibility[25,26]. As a result, efforts to develop complementary techniques for assaying molecular properties are ongoing. A recent example of such a development is the experimental polar surface area (EPSA) technique[27,28], which employs supercritical $CO_2$ fluid chromatography to assess the partitioning of an analyte between a polar stationary phase and a non-polar mobile phase. EPSA has been demonstrated as a highly efficient means for indirect detection of intramolecular hydrogen bonding[27]. Such intramolecular hydrogen bonds (IMHBs) are common features of organic molecules, playing a critical role in defining the conformations of small molecules[29], as well as the secondary structures of peptides and proteins[30]. With regard to drug design, introducing IMHBs has been put forward as a strategy to mask regions of higher polarity that might otherwise reduce cell permeability[28]. However, prior to the advent of the EPSA technique, identification of IMHBs, by either computation or experimentation, was challenging[27]. Thus, recent EPSA determinations of IMHB-containing molecules are quickly becoming a gold standard by which to test the ability of DMS to discern such features[31,32].

Unlike the majority of the molecules studied here, whose passive and fundamental physicochemical properties were our focus, the CRG species key attribute lies in understanding the reactivity trends of these molecules. These chemical reactivities are strongly influenced by their interaction potentials[33], making them good candidates for study via DMS. As added functional groups on drug molecules, CRGs react with specific nucleophilic residues in proteins, serving to silence enzymatic activity until protein re-synthesis can occur[33–37]. To date, dozens of CRG-modified drugs have been approved for treatment of hyperlipidemia, infectious diseases, and cancer[37]. These drugs contain electrophilic moieties such as carbamates, acetates, β-lactones, β-lactams, and acrylamides, and several reviews have described the efforts to develop covalent inhibitor therapeutics[34–37], including attempts to address-negative side-effects of these functional groups[38–40]. Given this considerable interest, there is an impetus towards a more detailed understanding of physicochemical and pharmacological properties of molecules of this type.

But, one question remains: How could the gas-phase microsolvation properties of a molecule, as measured by the DMS-MS technique, correlate so strongly to a molecule's physicochemical properties as measured in bulk solution? This begins by considering the critical measurements during a DMS experiment—the relationship between the separation voltage (SV) and optimal compensation voltage (CV) for transmission of the molecule through the DMS cell[5–8,41]. Specifically, the more strongly a molecule interacts with solvent molecules purposefully added inside the DMS cell, the more negative the CV value at a given SV. This CV/SV pairing encodes the molecule's interaction potential with the solvent system on a microscopic scale. However, variations in dynamic clustering behavior (and, therefore, differential mobility) across different molecule types/chemistries can render such a comparison ineffective. By taking a global view of the complete DMS behavior using a dispersion plot[3,7–10], one can qualitatively determine whether the molecule of interest exhibits strong-clustering (type A), weak-clustering (type B), or hard sphere collision (type C) interactions with the gaseous environment of the DMS cell[3,7–10]. Note, that in some cases the species of interest does not strictly follow one of these three common behavior types (see for example Figure S10 and Figure S11). In these cases the molecular interactions vary in a complex way as SV increases.

To ensure that the DMS-based assessment of molecular physicochemical properties is applicable across different chemistries, it is necessary to assess the properties of several molecular topologies jointly. To this end, we have combined the experimental results of our previous studies on ten 2-methylquinoline derivatives and twenty-two 2-methylquinolin-8-ol derivatives with the results from thirty-three IMHB-containing drug candidates (*vide infra*)[2,4]. We have also undertaken a parallel study of 24 acrylamide-based covalent reactive groups (CRGs) and have likewise included these species in a global analysis for all 89 small molecule drug candidates. We demonstrate that, with the aid of supervised machine learning (ML), the full range of DMS dispersion plot SV/CV data can be used to provide a quantitative means of accurately assessing a variety of molecular properties which are related to an analyte's interaction potential.

## Results and discussion

**Differential mobility spectrometry**. With regard to the IMHB-containing molecules, we hypothesize that those exhibiting an IMHB should interact with protic solvent vapor more weakly than an isomeric form of the same molecule that lacks an IMHB. This is due to the competition between intramolecular and intermolecular hydrogen bonding with solvent molecules inside the DMS cell. If so, this will manifest in DMS measurements as a comparatively positive CV shift for IMHB-containing isomers. To test this, 16 sets of isomeric molecules were selected for study where one of the isomers contained within a given set exhibited an IMHB as determined by EPSA measurements[27]. In total, 33 IMHB-containing molecules were studied; two of these (set A1) are shown in Fig. 1, with the rest provided in as Supplementary

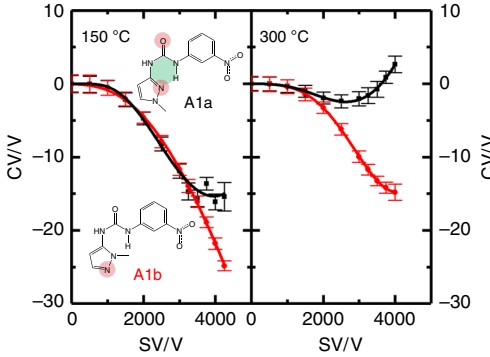

**Fig. 1** DMS dispersion plots of molecule set **A1**. Measurements were recorded in an $N_2$ environment seeded with methanol vapor (1.5% mole ratio) at $T = 150\,°C$ and $T = 300\,°C$. (Inset) Molecular topology **A**, set #1. Dispersion data for **A1a** and **A1b** are plotted in black and red, respectively. Compound **A1a** exhibits an IMHB (highlighted in green, whereas compound **A1b** does not). Protonation sites, as determined by DFT calculations, are highlighted in red. In **A1a**, the proton is shared between the carbonyl oxygen atom and ring nitrogen atom in the protonated form. Error bars ($2\sigma$) indicate the standard deviation of the Gaussian fit to the CV peak

Figures 7–22[10]. The IMHB-containing molecules are classified based on the IMHB data mining study of the Cambridge Structural Database published by Kuhn et al.[29] Here, four of the most common IMHB topologies were investigated, which we label as topologies **A–D**[10]. The numeric portion of the label (see, e.g., inset Fig. 1) indexes the isomeric set within a given topology, and the lower case letter labels a particular isomer within the set, with isomer **a** designating the species containing an IMHB[10].

The DMS dispersion plots for set **A1** shown in Fig. 1 were recorded in a $N_2$ environment that was modified with 1.5% (mole ratio) methanol vapor at DMS cell temperatures of 150 °C and 300 °C. The search for appropriate DMS conditions to effect isomer separation was a critical aspect of this study as isomer separation is indicative of differences in the isomers' relative interaction potentials. DMS experiments conducted using a pure $N_2$ environment, or a $N_2$ environment seeded with water, yielded hard sphere (Type C) behavior for all species[7,8,10], and hence no separation between isomers. A $N_2$ environment that was seeded with isopropyl alcohol vapor, on the other hand, yielded only strong-clustering (type A) behavior for all isomers[7,8,10]. Consequently, methanol—a protic solvent having properties intermediate of the previous two solvents tested—was chosen as the chemical modifier. Even so, at a temperature of 150 °C three sets of isomers (including set **A1**) could not be unambiguously separated into their isomeric components. However, at elevated temperatures the Gibbs' energy of binding for the ion-solvent clusters is reduced to the point where one can clearly distinguish between molecules exhibiting IMHBs and those not exhibiting IMHBs with DMS (Fig. 1)[7,8,10].

In all 16 IMHB sets studied, the isomer exhibiting an IMHB displayed DMS behavior indicating weaker ion-solvent interactions than did the non-IMHB-containing analogue(s). This is worth additional consideration as our study probes protonated forms of the targeted molecules, whereas the EPSA technique which identified the IMHB-containing molecules is presumed to probe the neutral compounds[27,28]. To support our experimental work, a detailed computational study was conducted in parallel[10]. In most cases, protonation was found to occur at or very near the site of IMHB formation proposed in the EPSA-based study[27]. Consequently, the principal site of solvent

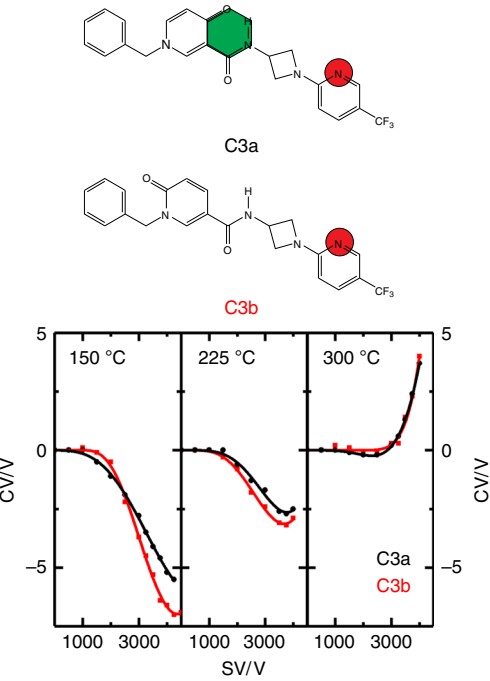

**Fig. 2** DMS dispersion plots of molecule set **C3**. Measurements were recorded in an $N_2$ environment seeded with methanol vapor (1.5% mole ratio) at $T = 150\,°C$, $T = 225\,°C$, and $T = 300\,°C$. The dispersion data of **C3a** are plotted in black and that of **C3b** is plotted in red. Molecule **C3a** exhibits an IMHB (highlighted in green), whereas molecule **C3b** does not. Protonation sites, as determined by DFT calculations, are highlighted in red. Errors are calculated as in Fig. 1, but are omitted for clarity

interaction is the same irrespective of whether the molecule is in a protonated gas-phase state or in a neutral condensed phase state. We have made this same observation in previous work[2]. In four cases (sets C2, C3, C4, and C5), the site of protonation was comparatively distal to the site of IMHB formation[10]. In these instances, as one might expect, the DMS behaviors of isomers were relatively similar and separation was more difficult to affect. We hypothesize that, as the site of protonation (and hence solvent interaction) is relatively distant from the site of IMHB formation in these species, it is necessary that the degree of solvent clustering for these molecules be relatively large to ensure that the hydrogen-bonding network of the protic solvent can interact with the site of IMHB formation. Experimentally, there are two ways to test this hypothesis; one can either vary the partial pressure of solvent vapor to influence the size of ion-solvent clusters formed during the low-field portion of the SV duty cycle, or one can vary the temperature of the gas to achieve this same outcome (i.e., vary the Gibbs' energy of solvent binding). Figure 2 shows the dispersion plots recorded for the C3 isomer set at temperatures of 150 °C, 225 °C, and 300 °C for an $N_2$ environment that was modified with 1.5% methanol vapor. At 300 °C, where the size of the ion-solvent clusters is expected to be relatively small, both isomers exhibit nearly identical dispersion plots. However, as the temperature of the collision gas is reduced, and the average size of the solvent clusters increases, we observed improved separation between the two isomeric species. Note also that the isomer which exhibits the IMHB (C3a) displays a more weakly clustering behavior than the non-IMHB isomer (C3b).

In addition to the 33 IMHB-containing molecules, 24 acrylamide CRGs were studied, which were selected from an earlier

examination of the intrinsic reactivities of these species[33]. These molecules can be classified into four different molecular topologies: (I) unsubstituted aromatic derivatives, (II) substituted aromatic derivatives, (III) substituted derivatives which do not contain heteroatoms, and (IV) substituted derivatives which contain heteroatoms. A more detailed description of these four structural motifs and the DFT-optimized geometries of all 24 derivatives are available as Supplementary Data[10]. In the case of the acrylamide CRGs, a $N_2$ environment (1 atm, 150 °C) that was seeded with 1.5% isopropyl alcohol vapor provided the widest distribution of analyte behavior (i.e., a distribution of type A, B, and C behavior). The dispersion plots for these species and all others included in this study are provided as Supplementary Figures 23-46[10].

**Analysis via supervised machine learning**. Although it is satisfying that DMS measurements can, for example, discriminate between isomeric species that do or do not contain IMHBs, the ultimate goal of this work is to demonstrate the use of DMS as a tool for quantitatively determining condensed phase physicochemical properties of molecules. To this end, we have introduced supervised machine learning (ML) as a means of inferring the relationships/correlations between molecular properties and DMS behavior. Preliminary efforts involved a survey of a few ML models[42]. Model selection is based on finding a compromise between bias and variance while avoiding over-fitting. Briefly, bias in a ML model can result from preselecting a functional description for your data, while high variance in a ML model will yield very different results depending on how data are partitioned into training and test sets. Over-fitting a model leads to poor applicability outside the range of molecules used for training. Decision Tree[43]-based models are unbiased learners and in particular Random Forest Regression[44] yields results with low variance and shows a low susceptibility to over-fitting. Further details of our ML study are provided in Supplementary Method[10].

In our previous work on methylquinoline-8-ol derivatives, we were able to identify a weak correlation between the turn-around point in the dispersion plot of specifically type B systems and cell permeabilities[2,45]. As other types of DMS behavior do not show such a turn-around point, generalizing these findings across all molecular types is not trivial. To remedy this shortcoming, machine learning was used to infer relationships between physicochemical properties and the entire DMS SV/CV datasets. (i.e. data corresponding to the entire dispersion plot are utilized). Specifically, we train our ML model using Random Forest with 500 random decision trees. In general, we were able to observe a relatively strong correlation between the observed DMS behavior of drug candidates and physicochemical properties. This is in line with expectation since many properties are to a large extent dependent on molecular size and on the solvent interactions, i.e., propensity for desolvation and transport across the hydrophobic lipid bilayer[13–15]. It therefore stands to reason that including the molecular collision cross section (CCS) as a descriptor in the ML database should improve the accuracy and precision of the ML model. To do this, CCSs were calculated with the updated MobCal code reported by Campuzano et al.[46] using DFT-optimized molecular structures[10,47,48]. A comparison of ML fits for cell permeability that include only DMS data with analogous ML fits that include DMS data and calculated CCSs are provided in Supplementary Methods, Supplementary Figures 47–50[10]. Although the fit containing only the DMS data exhibit a strong correlation with the experimentally determined RRCK rates of cell permeability ($R^2 = 0.889$), inclusion of the CCSs for the protonated drug molecules results in a significant improvement ($R^2 = 0.989$). On the other hand, if only CCS values are used to predict permeability we observe no significant correlation[10].

To investigate the performance of our model outside the training data, we use an iterative leave-one-out method whereby one molecule is selected to be left out of the training set and the resultant ML model is used to predict the properties for only this molecule. The selected molecule is then exchanged with another from the set, ML training is repeated, properties are calculated for the new molecule, and so on. This methodology is akin to an $N$-fold cross-validation procedure (where $N$ is the number of molecules in the set of compounds). Figure 3b, d shows fits for CCS and pKb, respectively. To monitor the evolution of test set and training set error as a function of training size for each property set, learning curves are created. To create the learning curves, the data is split into 10-folds and the ML model is trained using between 1 to 6-folds and tested on the remaining folds. Test/train folds are iterated through the dataset to determine errors for all molecules. Figure 3a, c shows learning curves for CCS and pKb, respectively. Fits as well as learning curves for LogD[49,50], EPSA[27,28], and cell permeability are provided in Supplementary Methods, Supplementary Figures 51–56[10]. The CCS fits (Fig. 3a, b) use only DMS data for the ML input, whereas the pKb fits (Fig. 3c, d) employ DMS data and calculated CCS values as inputs. In general, ML predictions for test set molecular properties correlate strongly with those derived experimentally by other means. Moreover, the error in test set predictions tends toward the error in the original measurements as database size increases. Additionally, learning curves show that testing on a larger dataset is required to confirm that these errors converge. Note that many of the properties determined using this methodology are either difficult to accurately compute from first principles, or ab initio methods for computing the property in question are yet to be developed. A brief description of computational approaches with comparison to our calculations is given as a Supplementary Discussion[10].

In the case of CCS fits, absolute errors tended to increase with increasing molecular size. However, the percent error in CCS was relatively constant across the dataset. For the 89 species studied, we find an average error of ca. 1.5% (Fig. 3b) for determining CCS via the ML model[10]. Note that the fits shown in Fig. 3 include data for multiple small molecule drugs of varying size and variable chemistry. This suggests that ML models constructed from DMS data may be broadly applicable across a wide range of molecular topologies. Fits to pKb (52 molecules, Fig. 3d), EPSA (43 molecules)[10], and logD (34 molecules)[10] involve a smaller subset of the ML database for which these condensed phase properties are available. Regardless of this limitation, correlation with DMS data are relatively strong. Because of this, we are optimistic that DMS behavior can also be used to predict these properties once a more extensive ML training database is constructed.

In summary, through application of supervised ML and consideration of molecular size (viz. CCS), one can correlate dynamic gas-phase clustering behavior within the DMS environment with condensed phase physicochemical properties. This suggests that DMS behavior is affected by the molecular interaction potentials of the analyte and collision gas within the DMS cell. This is demonstrated by, e.g., the fact that DMS can separate isomeric species based on the presence or absence of an IMHB; species that exhibit IMHBs display weaker ion-solvent clustering behavior than do the non-IMHB-containing analogues. The strong correlations observed between the DMS data and molecular physicochemical properties is appealing as it suggests that DMS might find use as a fast and accurate means of simultaneously screening molecular properties such as polar surface area, distribution coefficients, solubility, and cell permeability. This, in conjunction with the fact that DMS measurements can be conducted in minutes using only nanograms of sample, makes properties determination via DMS an attractive

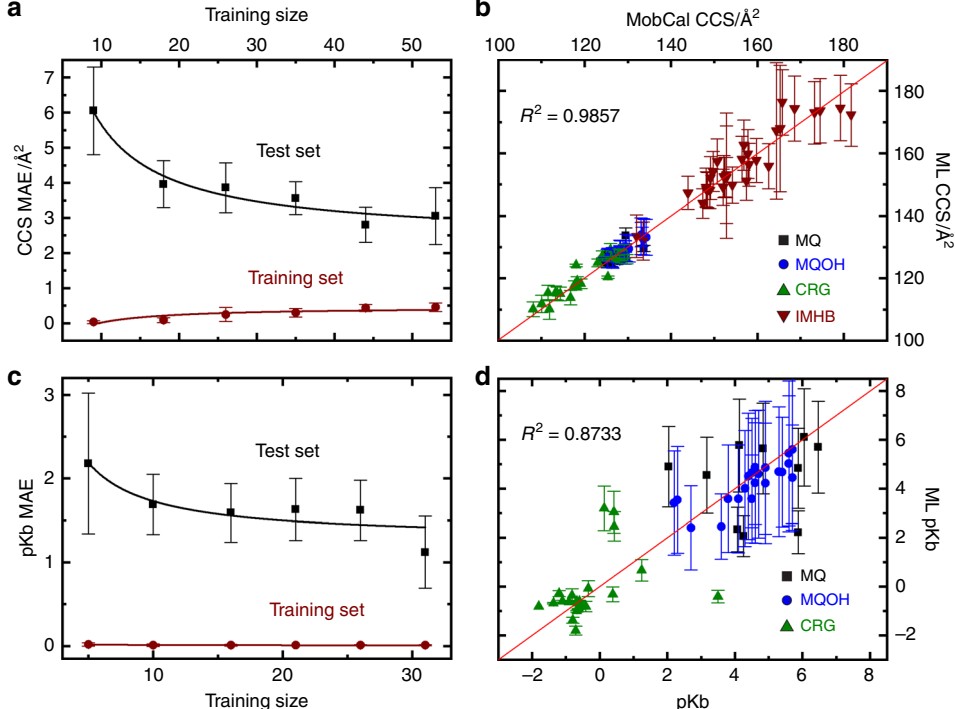

**Fig. 3** Random forest regression ML fits for CCS and pKb along with learning curves. Learning curves for CCS **a** and pKb **c** show the evolution of mean absolute error for test set (black) and training set (red) as the size of the ML training set is increased. MAE for test set is shown in black while that for training set is shown in red. Random forest results for CCS and pKb are shown in **b** and **d**, respectively. Individual points are colored based upon the molecular set as indicated in the figure. The red solid line in **b** and **d** is a plot of $y = x$. Error bars ($2\sigma$) are calculated from the standard deviation of the fitted parameter

possibility for drug discovery. To advance this technology, it is necessary to develop an extensive database of DMS behaviors and properties for a wide range of molecules. Our work is ongoing in this regard.

## Methods

**Differential mobility spectrometry details.** Drug candidates were provided by Pfizer Inc. and were used without further purification. HPLC-grade methanol (Caledon Laboratory Chemicals, Georgetown, ON) was also used without further purification. Distilled deionized water (18 MΩ) was produced in-house using a Millipore (Billerica, MA) Integral 10 water purification system. In each experiment, an analyte solution (10 ng/mL) was infused at a rate of 15 μL/min into DMS-MS instrument via an ESI source operating at 5.5 kV, with a source temperature of 300 °C, nebulizing gas pressure of 20 psi, and auxiliary gas pressure of 20 psi. The DMS cell (SelexION™, SCIEX, Concord, ON) system was mounted on a 5500 QTRAP® system (SCIEX), between a TurboV™ ESI source and the mass spectrometer's sampling orifice. Nitrogen was used as the curtain gas (3.5 L/min), throttle gas (0 or 0.7 L/min), and target gas (~3 mTorr) for the MS/MS experiments.

**Computational details.** To determine the most stable molecular geometries, a custom-written basin hopping (BH) search algorithm was used to map the potential energy surface (PES) of each drug candidate. The BH algorithm has been described in detail elsewhere[10,51]. Briefly, we first identified the most likely site(s) of protonated for each molecule by generating all possible tautomers/protomers and optimizing them individually at the B3LYP/LANL2DZ level of theory. For the most stable structures, atomic partial charges were calculated using the ChelpG partition scheme[52]. The cluster PES was then modeled using the Universal Force Field[53]. To search the PES, the dihedral angles associated with single bonds were randomly distorted by $-5° \geq \theta \geq +5°$ at each iteration of the BH code. In total, ~20,000 geometries were sampled for each molecule. Unique structures were identified based on zero-point corrected energy and geometry. Unique structures were then carried forward for geometry optimization at the PM7 level of theory for all molecules. To verify that calculations at the PM7 level produced appropriate geometries, the following subset of molecules were also geometry optimized at the B3LYP/6-311++G(d,p) level of theory: A1a, A1b, A4a, A4b, B1a, B1b, C4a, C4b, C4c, D1a, D1b.

For all molecules in the above subset, PM7 calculations produced the same global minimum structure and the same relative energy order of tautomers/protomers as did the B3LYP/6-311++G(d,p) calculations. In addition to geometry optimizations, normal mode analyses were also conducted to ensure that each structure was a local minimum on the PES.

**Machine learning details.** Supervised machine learning was conducted using the Orange3 Python package[54]. The labeled molecules were entered into a database which also included DMS data and calculated collision cross-sections (CCSs) as determined using the updated MobCal code reported by Campuzano et al.[46]. The Random Forest Regression employed 500 random decision trees and determined molecular properties via leave-one-out. This procedure trains the ML model using all molecules in the data except one, and then determines the property for the omitted molecule. This procedure is repeated for every molecule in the set. A series of correlation plots showing how the random forest (RF) model improves with additional data are shown in Supplementary Figures 47–50.

To produce learning curves, the database is split in to training and test sets. A ML model is produced from the training set and the mean absolute errors for the training and test sets are determined by applying the model to both. The plots provided in Fig. 3a, c, and Supplementary Figures 51–53 are produced by splitting the database, by molecular identity, into training sets consisting of 10, 20, 30, 40, 50, and 60% of the full database. The test set consists of data for the remaining molecules. The species selected for training are chosen randomly from the full database. Furthermore, for each of the fractional splits described above, the mean absolute error for each set is calculated using 10 different randomly chosen test sets and the reported error is determined as the average. i.e. 10 different test/train splits are constructed for each of the 10, 20, 30, 40, 50, and 60% cases, meaning in total the database is split 60 different ways to produce the learning curves given below. Error bars are calculated as one standard deviation of the average mean absolute error.

## Data availability

All relevant data for this publication are included in the manuscript and/or as supplementary material. In addition, the database is included as a data file with accompanying description. The custom-written basin hopping code is available from the authors upon request.

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

## Acknowledgements

We gratefully acknowledge support from the SHARCNET consortium of Compute Canada and from NSERC of Canada via a Discovery Grant and a Collaborative Research and Development (CRD) grant (#490885). We also thank the Ontario Centres of Excellence (OCE) for financial support through a Voucher for Innovation and Productivity II grant

(#25050). Furthermore, we would like to acknowledge Mr. Mark Zanon, Mr. Johnny Steffen, and Ms. Dalia Nasser for aid with early computational studies, and Prof. Roger Melko (Waterloo) for helpful discussions regarding machine learning.

## Author contributions

S.W.C.W., A.A., J.M.P. and J.C. contributed equally to the study. S.W.C.W., A.A. and J.M.P. performed DMS measurements for the various molecular classes. J.C. conducted the ML modeling and analysis, and prepared the final version of the manuscript. W.S.H. designed the project, initiated the original ML workflow and wrote the original manuscript. J.L.C. and J.S.J. aided WSH in direction of the science. J.L.C., J.S.J., C.L., J.C.Y.L.B., J.M. and G.H.G. provided helpful insights during data analysis.

## Additional information

**Competing interests:** The authors declare no competing interests.

