## [Peer Review File · Nature Communications]

Reviewers' comments:

Reviewer #1 (Remarks to the Author):

See attached.

Reviewer #2 (Remarks to the Author):

The authors study a series of drug-like molecules with differential mobility spectrometry (DMS). In this case the molecule interacts with an inert buffer gas and possibly solvent molecules. By using the DMS dispersion plots, the authors then construct a statistical model that aims to correlate DMS data to molecular properties in the condensed phase (cell permeability, EPSA, PKb, etc.). Neither the DMS analysis (which was already done in Ref. 2) nor the machine learning (ML) part (which has also been used extensively for molecular properties) are novel enough for Nature Communications. One could argue that the combination of DMS analysis with ML is novel, but in my opinion this is not the case.

My main criticisms to the manuscript are:

(1) The presentation is very technical and hard to follow. I do not see what is the major advance compared to the existing literature on chemoinformatics. The employed ML techniques are rather standard by now. There are way too much jargon and acronyms for the wide readership of Nature Communications.

(2) I fail to see fundamental advances or novel interpretations that would allow to assess the broad applicability of the presented approach. There are insufficient details on the

employed ML procedure, maximum errors, etc.

(3) A more detailed analysis for the uncovered correlations between DMS data and condensed phase physicochemical properties. Why do these correlations hold and for which systems? When do we expect them to fail? In principle, random forests allows to extract insights into such questions. Without such analysis and interpretation, I don't feel that I learn anything from this study.

Reviewer #3 (Remarks to the Author):

See attached.

The focus of this manuscript is the use of experimental data to predict a range of condensed phase molecular properties. The development of new assay techniques for use in machine learning approaches has value. However, the current level of detail regarding the machine learning in this manuscript is insufficient to warrant publication.

1. The authors describe using k-nearest neighbors, decision trees, adaptive boosting and random forests. They also state that the best model, based upon R² values is the random forest. What were the accuracies for the other approaches? Why a random forest, other than increased accuracy? Do the data type and number suggest the type of model most appropriate?
2. On line 236, the authors state that increasing the ML data base size reduced errors. This prompts several questions. Did the authors create a learning curve? Were the new data added at once? Was the model accuracy calculated after addition of either new temperature data or additional modifier conditions?
3. The authors generally provide R² values for models using either DMS only or DMS and CCS data. Was a model created using CCSs only? Which feature is more important?
4. The data used in this manuscript should be included as an electronic supplementary file.
5. The structures provided in Figure 1 and Figure 2 do not match the topologies in the Supplementary data. The labels in Figure 1 should be C1a and C1b NOT A1a and A1b. The labels in Figure 2 should be A3a and A3b NOT C3a and C3b.
6. Many of the properties that can be predicted in this study can also be directly calculated using standard cheminformatics software. How do the predictions of the models created here compare to normal computational routes (ex: polar surface areas)?

Reviewer #1

The manuscript describes a study wherein the authors propose a novel and original approach for the determination of molecular properties in the context of drug discovery. A combination of differential mobility spectrometry (DMS) integrated to mass spectrometry, and quantum chemistry is used to derive structural parameters of isomers of small molecules.

This work relies on previous findings of J.L. Campbell, W. S. Hopkins and coworkers who have shown that ion-molecule binding energy, correlates with parameters derived from DMS dispersion plots [see reference #2]. The principal added values of this manuscript is that a fairly large set of molecules has been studied, and that a supervised machine learning treatment has been using to « treat the differential mobility spectrometry – mass spectrometry (DMS-MS) data for ten topological classes of drug candidates to demonstrate that dynamic gas phase clustering behavior can be used to predict quantitatively the candidates ».

Overall, the manuscript is well written and could be of interest to a large audience. However, I have two concerns.

First, I think that the analysis performed using the machine learning (ML) scheme would deserve more a more detailed discussion. From Figure 3, it is clear that a much better correlation between ML and experimental cell permeability can be found when CCS are used for the ML process. It would have been to have a critical discussion on this important point.

My comments are thus essentially related to the DMS dispersion plots and quantum chemical calculation results.

I would thus recommend publication of this manuscript subject to minor revision, and would like to stress that I am unable to provide a critical review on the Machine Learning section.

As far as the DMS reasoning is concerned, my comments are essentially related to some statements made by the authors (lines 150-159) :

« This is worth additional consideration since our study probes protonated forms of the targeted molecules, whereas the EPSA technique which identified the IMHB-containing molecules is presumed to probe the neutral compounds.[27, 28] To support our experimental work, a detailed computational study was conducted in parallel.[42] In most cases, protonation was found to occur at or very near the site of IMHB formation proposed in the EPSA-based study.[27] Consequently, the principal site of solvent interaction is the same irrespective of whether the molecule is in a protonated gas phase state or in a neutral condensed phase state. We have made this same observation in previous work.[2] In three cases (sets C2, C3, and C5), the site of protonation was comparatively distal to the site of IMHB formation.[42] »

After a careful analysis of the data provided in the manuscript and supplementary information section, here are my comments :

- First of all, it would be interesting to explain how the DMS parameter (SV at minimum CV) is derived for the multiple ions showing a DMS dispersion plot of type C at 300°C.
- The data reported in Figure 1 do not correspond to molecule set A1 but rather C1.
- Conversely, data reported in Figure 2 do not correspond to molecule set C3 but rather A3.
- The DMS dispersion voltage data recorded for molecules B3b (Figure S9) and B4b (Figure S10) deserve a specific comment.
- As stated (lines 158-159), the site of protonation can be comparatively distal to the site of IMHB formation. In such case, the charge should be available for microsolvation, and one would expect the corresponding DMS dispersion voltage plot to correspond to the so-called « type A » or « type B » ion. An inspection of the results (supplementary information) shows that protonation occurs distal to the site of IMHB formation not only in the cases of sets C2, C3, and C5, but also in the cases of A2, A3, A4, and A5. The sentence (line 158-159) should be corrected.
- Previous works of the authors [reference #2, for example] clearly demonstrates that when the charge site of the ion is available for solvation, the type A or B DMS dispersion plot is observed. This could be a

consequence of the high temperature (300°C) of the DMS device, but I found strange that protonated A2b, A3a, and A3b behave as hard sphere (type C ions) since the positive charge is well available for solvation.

- In order to verify that calculations at the PM7 level produced appropriate geometries, the subset of molecules were also geometry optimized at the B3LYP/6-311++G(d,p) level of theory (lines 660-662 in SI). I understand that giving too much details would be out of the scope of the present study, but looking at the structures and corresponding dispersion plots of set A molecules, I have some questions. First, considering that A2a and A2b dispersion plots are similar, one would think that they have similar structures. Looking at the PM7 results, it does not seem to be the case : a keto group is protonated in the case of A2a, while a tertiary amine is protonated in the case of A2b. Second, the intramolecular hydrogen bonding (IMHB) changes upon protonation of A1a, which is expected since an ionic hydrogen bond is stronger than a neutral one. I was thus wondering why such a switch in hydrogen bonding does not occur for A2a ?

I thus think that A2 and A3 isomers deserve more attention.

Response to Reviewer 1

Reviewer 1 wrote:

"Overall, the manuscript is well written and could be of interest to a large audience. However, I have two concerns. First, I think that the analysis performed using the machine learning (ML) scheme would deserve more a more detailed discussion. From Figure 3, it is clear that a much better correlation between ML and experimental cell permeability can be found when CCS are used for the ML process. It would have been to have a critical discussion on this important point."

Thank you for the constructive comments. We have now included a brief discussion of the Random Forest Regression technique in the manuscript and further detailed discussion in the supplementary material. The supplementary material has been expanded to include learning curves and a discussion of expected errors. Additionally, machine learning fits using only CCS data have been included in supplementary material for comparison. The results are briefly mentioned in the main text as well.

My comments are thus essentially related to the DMS dispersion plots and quantum chemical calculation results. I would thus recommend publication of this manuscript subject to minor revision, and would like to stress that I am unable to provide a critical review on the Machine Learning section. As far as the DMS reasoning is concerned, my comments are essentially related to some statements made by the authors (lines 150-159) :

« This is worth additional consideration since our study probes protonated forms of the targeted molecules, whereas the EPSA technique which identified the IMHB-containing molecules is presumed to probe the neutral compounds.[27, 28] To support our experimental work, a detailed computational study was conducted in parallel.[42] In most cases, protonation was found to occur at or very near the site of IMHB formation proposed in the EPSA-based study.[27] Consequently, the principal site of solvent interaction is the same irrespective of whether the molecule is in a protonated gas phase state or in a neutral condensed phase state. We have made this same observation in previous work.[2] In three cases (sets C2, C3, and C5), the site of protonation was comparatively distal to the site of IMHB formation.[42] »

After a careful analysis of the data provided in the manuscript and supplementary information section, here are my comments:

- First of all, it would be interesting to explain how the DMS parameter (SV at minimum CV) is derived for the multiple ions showing a DMS dispersion plot of type C at 300°C.

Unlike in our earlier study [Liu C, *et al.* Assessing Physicochemical Properties of Drug Molecules via Microsolvation Measurements with Differential Mobility Spectrometry. *ACS Cent Sci* 3, 101-109 (2017)], here we do not limit ourselves to using only SV at minimum CV. In this study we apply machine learning to make use of the full SV/CV dataset for each molecule in the ML database. This allows us to treat all molecules types (Types A and C included), rather than just Type B molecules. To clarify this further, we have expanded the description in Section IIIa.

- The data reported in Figure 1 do not correspond to molecule set A1 but rather C1.

We apologize for the mislabeled figure. The labelling throughout the supplementary material has been updated.

- Conversely, data reported in Figure 2 do not correspond to molecule set C3 but rather A3.

We apologize for the mislabeled figure. The labelling throughout the supplementary material has been updated.

- The DMS dispersion voltage data recorded for molecules B3b (Figure S9) and B4b (Figure S10) deserve a specific comment.

We agree. A sentence has been included in Section II which directs the reader to these figures which also include a brief comment on the different behavior.

- As stated (lines 158-159), the site of protonation can be comparatively distal to the site of IMHB formation. In such case, the charge should be available for microsolvation, and one would expect the corresponding DMS dispersion voltage plot to correspond to the so-called « type A » or « type B » ion. An inspection of the results (supplementary information) shows that protonation occurs distal to the site of IMHB formation not only in the cases of sets C2, C3, and C5, but also in the cases of A2, A3, A4, and A5. The sentence (line 158-159) should be corrected.

You are correct. This was again due to the mislabeled Molecule sets, as in Figure 1 and Figure 2. Additionally C4 should have been in the list. The sentence now refers to C2, C3, C4, and C5.

- Previous works of the authors [reference #2, for example] clearly demonstrates that when the charge site of the ion is available for solvation, the type A or B DMS dispersion plot is observed. This could be a consequence of the high temperature (300°C) of the DMS device, but I found strange that protonated A2b, A3a, and A3b behave as hard sphere (type C ions) since the positive charge is well available for solvation.

In these cases, the chemical understanding is being clouded by the two-dimensional stick structures. In all three of these cases the actual 3D geometric structure, as determined by B3LYP/6-311++G(d,p) calculations, is folded such that the positive charge is more protected than is inherently obvious from the stick figures. The 3D structures are provided in the supplementary material.

- In order to verify that calculations at the PM7 level produced appropriate geometries, the subset of molecules were also geometry optimized at the B3LYP/6-311++G(d,p) level of theory (lines 660-662 in SI). I understand that giving too much details would be out of the scope of the present study but looking at the structures and corresponding dispersion plots of set A molecules, I have some questions. First, considering that A2a and A2b dispersion plots are similar, one would think that they have similar structures. Looking at the PM7 results, it does not seem to be the case: a keto group is protonated in the case of A2a, while a tertiary amine is protonated in the case of A2b. Second, the intramolecular hydrogen bonding (IMHB) changes upon protonation of A1a, which is expected since an ionic hydrogen bond is stronger than a neutral one. I was thus wondering why such a switch in hydrogen bonding does not occur for A2a? I thus think that A2 and A3 isomers deserve more attention.

In the case of A2a/A2b (now C2a/C2b) DMS separation is more obvious in DMS measurements performed at 150 °C as compared with 300 °C, presumably due to slight differences in structure and interaction. For A2b (now C2b), protonation on the tertiary amine is shared with the carbonyl oxygen. As the reviewer points out, the switch in hydrogen bonding is possible for A2a (now C2a), but it results in a higher energy isomer (+0.17 eV at the PM7 level of theory).

Response to Reviewer 2

Reviewer 2 wrote:

“The authors study a series of drug-like molecules with differential mobility spectrometry (DMS). In this case the molecule interacts with an inert buffer gas and possibly solvent molecules. By using the DMS dispersion plots, the authors then construct a statistical model that aims to correlate DMS data to molecular properties in the condensed phase (cell permeability, EPSA, PKb, etc.). Neither the DMS

analysis (which was already done in Ref. 2) nor the machine learning (ML) part (which has also been used extensively for molecular properties) are novel enough for Nature Communications. One could argue that the combination of DMS analysis with ML is novel, but in my opinion, this is not the case.”

It is true that individually/separately DMS and the ML algorithms employed are not novel. However, our application of ML to the DMS dataset enables properties measurements that have not been possible in the previous 40-year history of the DMS technique.

In reference 2 [Liu C, et al., *ACS Cent Sci* 3, 101-109 (2017)], correlations between DMS measurements and physicochemical properties could only be determined for compounds which display specifically Type B behavior (using measurement of the SV value at the minimum CV value). Moreover, the observed trends/correlations were only apparent within a series of closely related compounds (quinoline derivatives); compounds of differing structure / chemistry could not be treated simultaneously.

The application of machine learning allows us to treat all compounds, regardless of structure, chemistry, or DMS-behaviour (viz. Type A, B, or C). This substantially increases the applicability and power of using DMS to assess a variety of molecular properties simultaneously, in minutes, using only pico-to-nanograms of sample. Many of the DMS / property correlations are reported here for the first time. In fact, many in the field did not think that it was possible to extract these properties from DMS data. This is only possible through treatment with machine learning. To further clarify this point, Section IIIa has been expanded.

My main criticisms to the manuscript are:

(1) The presentation is very technical and hard to follow. I do not see what is the major advance compared to the existing literature on chemoinformatics. The employed ML techniques are rather standard by now. There are way too much jargon and acronyms for the wide readership of Nature Communications.

The goal of this work is not to develop new data processing algorithms, but rather to apply established ML methods to treat DMS data. We do this here for the first time and demonstrate the determination of numerous molecular properties. The ability to measure the reported p-chem properties simultaneously, in minutes, with only pico-to-nanograms of sample represents a substantial advancement for properties measurement; with current state of the art, mg-to-g of sample and days-to-weeks of measurement time are required to determine the reported properties.

We have attempted to re-write the text to ease readability.

(2) I fail to see fundamental advances or novel interpretations that would allow to assess the broad applicability of the presented approach. There are insufficient details on the employed ML procedure, maximum errors, etc.

We have now expanded the description of the ML procedure in the manuscript, while attempting to exclude as much jargon / technical detail as possible for ease of readability. The added text includes a discussion of why the model was chosen and additional fits to assess applicability of the models for predictions outside the training set.

(3) A more detailed analysis for the uncovered correlations between DMS data and condensed phase physicochemical properties. Why do these correlations hold and for which systems? When do we expect them to fail? In principle, random forests allows to extract insights into such questions. Without such analysis and interpretation, I don't feel that I learn anything from this study.

The correlations we show are expected to hold for molecular properties which are influenced by molecular size and/or intermolecular interaction potentials (e.g., solubility, cell permeability). To this point, we show strong correlations for all ten topological systems investigated. We have also now included learning curves to estimate errors for predictions outside of the training sets. Currently, we are limited to small molecules with defined, relatively simple interaction potentials. We are in process of investigating larger, more complex systems.

Response to Reviewer 3:

The focus of this manuscript is the use of experimental data to predict a range of condensed phase molecular properties. The development of new assay techniques for use in machine learning approaches has value. However, the current level of detail regarding the machine learning in this manuscript is insufficient to warrant publication.

1. The authors describe using k-nearest neighbors, decision trees, adaptive boosting and random forests. They also state that the best model, based upon R² values is the random forest. What were the accuracies for the other approaches? Why a random forest, other than increased accuracy? Do the data type and number suggest the type of model most appropriate?

We have updated the discussion on machine learning algorithms to better describe why Random Forest was chosen. Additionally, machine learning fits using only CCS data have been included in the supplementary material for comparison. The results are highlighted in the main text as well.

2. On line 236, the authors that increasing the ML data base size reduced errors. This prompts several questions. Did the authors create a learning curve? Were the new data added at once? Was the model accuracy calculated after addition or either new temperature data or additional modifier conditions?

Learning curves have now been included as supplementary material and they are briefly mentioned in main text. These were constructed by splitting our data into training and test sets of varying sizes. We have attempted to clarify our procedure in the main text of the manuscript.

3. The authors generally provide R² values for models using either DMS only or DMS and CCS data. Was a model created using CCSs only? Which feature is more important?

ML fits for other p-chem properties were conducted using only CCS input data. In general, there is little-to-no correlation with CCS data alone (*viz.* DMS data is essential for good correlation). This suggests that the intermolecular interactions potentials (which influence DMS trajectories) is the more important feature. The fits are provided in supplementary material.

4. The data used in this manuscript should be included as an electronic supplementary file.

Data for the ML fits have been included as a Supplementary file.

5. The structures provided in Figure 1 and Figure 2 do not match the topologies in the Supplementary data. The labels in Figure 1 should be C1a and C1b NOT A1a and A1b. The labels in Figure 2 should be A3a and A3b NOT C3a and C3b.

We apologize for the mislabeled figure. The labelling has been fixed in the supplementary material to reflect the correct sets.

6. Many of the properties that can be predicted in this study can also be directly calculated using standard cheminformatics software. How do the predictions of the models created here compare to normal computational routes (ex: polar surface areas)?

The ML fit errors tend towards the measurement error of the property for which the ML model was constructed. A larger data set is required to verify whether the ML model errors do, in fact, converge to the errors in the measured properties.

Reviewer #1 (Remarks to the Author):

I have read carefully the responses to the three reviewers and the manuscript.

As said in my first review, the manuscript is well written and could be of interest to a large audience. I made two points:

- 1) I found the answers to my technical questions.
- 2) My main concern is still related to the Machine Learning section. As I said, I was unable to provide a critical review on this section, and I am still unable to make up my mind.

Reviewer #2 (Remarks to the Author):

I stand by my original decision that this manuscript is not appropriate for the broad readership of Nature Communications.

My main arguments were the rather technical nature of the presentation and the lack of coherence in tying the experimental data with the employed machine learning approach.

Upon reading the reply, my opinion is even further reinforced by the new discussion that the authors provided concerning their machine learning (ML) approach.

The determination of hyperparameters in machine learning is based on the statistically

rigorous idea of N-fold cross validation. Only fully cross-validated results on test sets should be shown in any study that employs machine learning. Many of the authors' conclusions are based on errors on the training set (see Figure 3 and its discussion). This is not rigorous and leads to far too optimistic results. This is in fact clear from Figure 4, where the performance on test set is far worse. There are many studies that demonstrate appropriate procedures for cross validation and testing. For actual "best practices" of using ML for chemical data, see <https://pubs.acs.org/doi/abs/10.1021/ct400195d>.

In summary, I have serious doubts about the statistical validity of the conclusions presented in this manuscript and don't think this work satisfies the most basic requirements for publication in Nature Communications.

Reviewer #3 (Remarks to the Author):

The authors' revisions have increased the clarity of the manuscript. The increased detail on the machine learning aspects of this work have greatly increased its utility to a wider audience.

The authors state that the ML errors 'tend towards the measurement error of the property for which the ML model was constructed' in response to a question about the differences between this ML work and standard cheminformatics approaches to calculating some of the same properties. This response compares the ML results to experimental details, which certainly has value. However, a comparison of how their ML-predicted values compare to other calculated values is where the utility of this work will be most clearly stated. The impact of this work is based upon the development of a better way to determine these values, as compared to both experimental values and competing computational work.

The authors describe model choice in section IIIa. The rationale for choosing a random forest approach are sound. However, the authors should report accuracies (or error rates) for the other approaches that they list.

The inclusion of the data in electronic form is welcome, as it will allow others to experiment with the techniques described in this manuscript. However, a plain language explanation of the data columns should be provided in a separate document.

Comments from reviewer 1:

I have read carefully the responses to the three reviewers and the manuscript. As said in my first review, the manuscript is well written and could be of interest to a large audience. I made two points:

- 1) I found the answers to my technical questions.
- 2) My main concern is still related to the Machine Learning section. As I said, I was unable to provide a critical review on this section, and I am still unable to make up my mind.

Comments from reviewer 2:

I stand by my original decision that this manuscript is not appropriate for the broad readership of Nature Communications. My main arguments were the rather technical nature of the presentation and the lack of coherence in tying the experimental data with the employed machine learning approach. Upon reading the reply, my opinion is even further reinforced by the new discussion that the authors provided concerning their machine learning (ML) approach.

Author's Response: Our manuscript demonstrates that one can quickly and accurately determine a suite of physicochemical properties simultaneously for small molecules by applying supervised machine learning to DMS data. These data can be acquired in minutes for multiple compounds, using only nanograms of relatively impure sample. This is a substantial improvement over existing p-chem properties measurement techniques and it demonstrates a new application of the DMS technology. We, and our co-authors at SCIEX and Pfizer, feel that this is appealing to the broad readership of Nature Communications.

The determination of hyperparameters in machine learning is based on the statistically rigorous idea of N-fold cross validation. Only fully cross-validated results on test sets should be shown in any study that employs machine learning. Many of the authors' conclusions are based on errors on the training set (see Figure 3 and its discussion). This is not rigorous and leads to far too optimistic results. This is in fact clear from Figure 4, where the performance on test set is far worse. There are many studies that demonstrate appropriate procedures for cross validation and testing. For actual "best practices" of using ML for chemical data, see <https://pubs.acs.org/doi/abs/10.1021/ct400195d>.

Author's Response: Review #2 points to Figure 4 and says that the performance of our ML model is far worse when tested against data that is outside of our training set (as is expected) and because of this (s)he thinks that our treatment is too optimistic. First, please let me point out that Figure 4 shows the *best and worst* correlations from our studies. In the case of collision cross section, our R^2 value drops from 0.99 to 0.97. We explicitly state this in the manuscript. However, even with this poorer correlation, our model predictions are still approaching the accuracy of calculated values and experimental methods used to acquire the pharmacokinetic data (see response to Review #3 below). Moreover, since we first submitted our manuscript to Nature Chemistry (ca. 8 months ago), we have continued in our development of this

methodology. The Figure shown below plots the correlation between our ML model and the collision cross sections (*i.e.*, molecular sizes) of 220 molecules, which range in size from small molecule drugs up to multiply charged peptide hormones (*viz.*, Angiotensin 3+). It is clear from this figure that our model holds up to larger data sets.

Reviewer #2 further states that only "fully validated" results should be presented and (s)he provides a reference for "best practices" in ML. The reference states that "In general a k value of 5 or 10 is recommended by Breiman and Spector". This (k = 5; 5-fold validation) is what we did in our original submission. Upon request by Reviewer #2 following his/her initial assessment, we created learning curves for our model to show error as a function of k (*i.e.*, data split). The last point in those curves, and the test set correlations that we show in Figure 4, are "leave-one-out" cross-validation error estimates (*i.e.*, k = 89). These are more computationally demanding but are expected to provide the best error estimates. To clarify further in the manuscript we have now included the following on page 10:

“This methodology is akin to an N-fold cross validation procedure (where N is the number of molecules in the set of compounds). Learning curves showing the evolution of test set errors as a function of N are provided in the supplementary material.”

Reviewer #2 also states that "Many of the authors' conclusions are based on errors on the training set (see Figure 3 and its discussion)". This is not correct. Please see Figure 4 and the discussion that begins on page 10 of our manuscript, which starts with "To investigate the performance of our model outside the training set...". We clearly assess model performance against a test set. Perhaps Reviewer #2 just missed/overlooked that this is the case. We now explicitly state the test set errors in the SI.

I understand that Reviewer #2 comes from an informatics background and that (s)he is used to dealing with data sets that have several thousand entries, so our data set of 89 molecules (ca. 500 SV/CV entries) is small in comparison. Nevertheless, we believe that our work demonstrates the

validity of our approach, which will continue to improve as more data is added to our data base. We are quite confident that our methodology is successful. As further evidence of this, one of our industry partners (SCIEX) is in process of developing a cloud-based online service for molecular property measurement based on our methodology.

In summary, I have serious doubts about the statistical validity of the conclusions presented in this manuscript and don't think this work satisfies the most basic requirements for publication in Nature Communications.

Comments from reviewer 3:

The authors' revisions have increased the clarity of the manuscript. The increased detail on the machine learning aspects of this work have greatly increased its utility to a wider audience. The authors state that the ML errors 'tend towards the measurement error of the property for which the ML model was constructed' in response to a question about the differences between this ML work and standard cheminformatics approaches to calculating some of the same properties. This response compares the ML results to experimental details, which certainly has value. However, a comparison of how their ML-predicted values compare to other calculated values is where the utility of this work will be most clearly stated.

Author's Response: It is certainly true that physicochemical properties are first calculated as a means of assaying potential drug candidates. Thus, a comparison between our DMS-ML work and ab initio predictions is of value. The ability to "measure" a suite of molecular properties simultaneously and quickly with a relatively small amount of impure sample is of equal (or greater) value. To address the concerns of the reviewer with regard to comparing our method with computed properties, we have added to following to the SI and have directed the reader to this in the main text of the manuscript:

In the manuscript we explicitly show comparison with calculated CCS values as well as measured pKb values. The expected error for our computed CCS values is $< 3\%$ as compared with experimentally determined CCS values. Thus the error of our ML treatment is within the expected error of the CCS calculations.

Attempts to calculate pKa use either first principle quantum mechanics, cheminformatics, or a combination of the two. In a recent study on a large data set (> 1000 pKa values) using first principle QM calculations and fitting experimental values [Bochevarov et al. JCTC 2016], pKa was predicted for most compounds to within a mean unsigned error of < 1.0 for molecules with pKa's in the range of 6 - 12. The error we obtain on our pKa determinations is on the order of 0.93 for 64 compounds which possess pKa values between 7.5 and 16. Admittedly, comparing our somewhat limited test set (64 compounds) with the Bochevarov study (> 1000 compounds) is optimistic, but our work does indicate the potential viability of our approach.

ML fits were also performed for logD, EPSA, and cell permeability (see Figures S50 – S52). For all molecules studied, we used measured values for these quantities which covered a range of -0.54 – 2.97 for logD, 51 – 125 for EPSA, and 0 – 41.1 for cell permeability. The mean absolute

errors for the fits on each of these properties are 0.47, 17, and 4.6 respectively. EPSA [Refsgaard et al. J. Med. Chem. 2005] and logD are experimental measurements devised in order to estimate cell permeability (through polarity and lipophilicity measurement, respectively). Permeability is known to strongly depend on molecular size, polarity, and lipophilicity [Guimaraes et al. J. Chem. Inf. Model 2012]. Therefore, EPSA and logD can provide qualitative measures of molecular permeability. As far as we are aware, theoretical calculations of EPSA have not yet been devised, but logD (or equivalently logP) calculations are possible. Calculation of logP follows a group based summation method [Klopman et al. J. Chem. Inf. Comput. 1994] which performs exceedingly well on the 1663 compounds tested. Standard errors were 0.58 for a logP values ranging from -4 to +7. This is superior to the performance of our DMS-ML database; however, we would like to (again) highlight the fact that to this point our method only employs a test set of 66 molecules. The learning curves that we calculate suggest that the performance of our method should improve with the addition of new compounds to our training set.

Direct prediction of cell permeability is much more difficult but is possible using molecular dynamics [Bennion et al. J. Phys. Chem B., 2017]. In their approach the authors compare against 18 compounds for which they can provide a qualitative measure of permeability (*i.e.*, impermeable, low permeability, or high permeability). Qualitatively, the method of Bennion performs well (only 2 of 18 molecules are misclassified), but quantitative accuracy is poor. Our ability to determine cell permeability to within ca. 10 % for most molecules tested is certainly significant, especially given our relatively small test/training set.

The impact of this work is based upon the development of a better way to determine these values, as compared to both experimental values and competing computational work. The authors describe model choice in section IIIa. The rationale for choosing a random forest approach are sound. However, the authors should report accuracies (or error rates) for the other approaches that they list.

Author's Response: We have included ML fits for Adaptive Boosting and k-nearest neighbors in the SI, along with fit error as represented by R^2 and MAE. For the most part these fits are worse than those obtained using the Random Forest model.

The inclusion of the data in electronic form is welcome, as it will allow others to experiment with the techniques described in this manuscript. However, a plain language explanation of the data columns should be provided in a separate document.

Author's Response: A separate document titled: "Description_of_ML_input_all.docx", which provides a concise description of each column in the table, has now been provided. Additionally, a small section at the end of this document describes briefly how the data is used for machine learning fits.

Reviewers' comments:

Reviewer #4 (Remarks to the Author):

Firstly a technical point. Ten trees is far too few for a random forest, I would expect to see at least 100 trees in order to obtain the benefits of the RF algorithm and converge the results with respect to the number of trees. Justifying the use of a relatively small number of trees would mean producing a version of your existing learning curves for one case, with respect to additional trees rather than additional molecules. However, unless CPU time is more of a constraint than I expect, it's easier just to go to a larger number of trees (I'd use 500 by default).

This manuscript really reads like a full paper that's been shoehorned into the format of a Communication. The Introduction is long and detailed, though definitely informative. Some important material has been stranded in the ESI, for example the Details on Supervised Machine Learning and also on Learning Curves. It would be good to see at least one or two of your learning curves in the main manuscript.

Figure 3 of the existing manuscript can be dispensed with. I consider training set fits to be largely irrelevant unless they are very poor, and therefore this figure can be safely omitted.

In line 164, you write "to affect" (meaning "to influence") in a context that suggests that "to effect" (meaning "to bring about") might be more suitable.

Reviewers' comments:

Reviewer #4 (Remarks to the Author):

Firstly a technical point. Ten trees is far too few for a random forest, I would expect to see at least 100 trees in order to obtain the benefits of the RF algorithm and converge the results with respect to the number of trees. Justifying the use of a relatively small number of trees would mean producing a version of your existing learning curves for one case, with respect to additional trees rather than additional molecules. However, unless CPU time is more of a constraint than I expect, it's easier just to go to a larger number of trees (I'd use 500 by default).

Author's Comment: We have replaced all plots in the manuscript and supplementary information with fits that use 500 decision trees.

This manuscript really reads like a full paper that's been shoehorned into the format of a Communication. The Introduction is long and detailed, though definitely informative. Some important material has been stranded in the ESI, for example the Details on Supervised Machine Learning and also on Learning Curves. It would be good to see at least one or two of your learning curves in the main manuscript.

Author's Comment: Figure 4 of the manuscript has been changed to Figure 3, and Figure 3 has been moved to the supplementary material. We have replaced training set fits in the new Figure 3 (originally Figure 4) with learning curves for the relevant properties (CCS and pKb).

Figure 3 of the existing manuscript can be dispensed with. I consider training set fits to be largely irrelevant unless they are very poor, and therefore this figure can be safely omitted.

Author's Comment: done.

In line 164, you write "to affect" (meaning "to influence") in a context that suggests that "to effect" (meaning "to bring about") might be more suitable.

Author's Comment: Agreed. This has been changed.

REVIEWERS' COMMENTS:

Reviewer #4 (Remarks to the Author):

The remaining issues have been dealt with and I have nothing further to raise.

Reviewer #4 (Remarks to the Author):

The remaining issues have been dealt with and I have nothing further to raise.

Author's Comment: Great. Thank you.